# The Effect of Phytonutrients in *Terminalia chebula* Retz. on Rumen Fermentation Efficiency, Nitrogen Utilization, and Protozoal Population in Goats

**DOI:** 10.3390/ani12162022

**Published:** 2022-08-10

**Authors:** Pongsatorn Gunun, Anusorn Cherdthong, Pichad Khejornsart, Metha Wanapat, Sineenart Polyorach, Sungchhang Kang, Walailuck Kaewwongsa, Nirawan Gunun

**Affiliations:** 1Department of Animal Science, Faculty of Natural Resources, Rajamangala University of Technology Isan, Sakon Nakhon Campus, Phangkhon, Sakon Nakhon 47160, Thailand; 2Tropical Feed Resources Research and Development Center (TROFREC), Department of Animal Science, Faculty of Agriculture, Khon Kaen University, Khon Kaen 40002, Thailand; 3Agro-Bioresources, Faculty of Natural Resources and Agro-Industry, Kasetsart University, Chalermphakiat Sakon Nakhon Campus, Sakon Nakhon 47000, Thailand; 4Department of Animal Production Technology and Fisheries, Faculty of Agricultural Technology, King Mongkut’s Institute of Technology Ladkrabang, Bangkok 10520, Thailand; 5National Institute of Education, Phnom Penh 268, Cambodia; 6Department of Animal Science, Faculty of Technology, Udon Thani Rajabhat University, Udon Thani 41000, Thailand

**Keywords:** phytonutrients, *Terminalia chebula* Retz., rumen fermentation, nitrogen utilization, protozoal population

## Abstract

**Simple Summary:**

The manipulation of ruminal ecology and fermentation to improve feed efficiency and reduce nitrogen excretion in ruminants is a pertinent goal for animal nutritionists. The fruits of *Terminalia chebula* Retz. are a rich source of phytonutrient compounds, which are attractive as rumen modifiers since these compounds are natural substances. The purpose of this study is to determine the effect of *Terminalia*
*chebula* meal (TCM) supplementation on digestibility, rumen fermentation, nitrogen utilization, and protozoal population in goats. The findings indicated that the inclusion of TCM could improve rumen fermentation efficiency and N balance while reducing the protozoal population. As a result, TCM could be used as a plant source for the rumen enhancement of small ruminants.

**Abstract:**

The aim of this study was to investigate the effect of *Terminalia chebula* meal (TCM) supplementation on digestibility, rumen fermentation, nitrogen utilization, and protozoal population in goats. Eight goats with an initial body weight (BW) of 13 ± 3.0 kg were randomly assigned according to a double 4 × 4 Latin square design to receive different levels of TCM supplementation at 0, 8, 16, and 24 g/kg of total dry matter (DM) intake, respectively. The goats were fed with concentrate diets at 13 g/kg BW, while rice straw was used as a roughage source, fed *ad libitum*. The results revealed that the feed intake and the apparent digestibility of DM, organic matter (OM), neutral detergent fiber (NDF), and acid detergent fiber (ADF) were similar among the treatments (*p* > 0.05). However, crude protein (CP) digestibility decreased significantly (*p* < 0.05) when supplemented with TCM at 24 g/kg of total DM intake (*p* < 0.05). The addition of TCM did not change the ruminal pH and blood urea nitrogen concentrations (*p* > 0.05), whereas the concentration of NH_3_-N at 4 h post feeding was reduced with the inclusion of TCM at 16 and 24 g/kg of total DM intake. The total numbers of bacteria were enhanced by the addition of TCM, while the protozoal population, in both entodiniomorph and holotrich, was reduced (*p* < 0.05). The supplementation of TCM did not change the concentration of total volatile fatty acids (TVFA), acetic acid, or butyric acid, while the propionic acid concentration at 4-h post feeding increased significantly, especially when supplemented at 16 g/kg of total DM intake (*p* < 0.05. In addition, urinary nitrogen (N) excretion decreased, while fecal N excretion, N absorption, N retention, and the proportion of N retention to N intake increased with the inclusion of TCM at all levels. In summary, the inclusion of TCM could improve rumen fermentation efficiency and N balance without having an adverse effect on feed intake, nutrient digestibility, and rumen ecology; however, the protozoal population decreased. Therefore, this study suggests that TCM (16 g/kg of total DM intake) could be used as a plant source for rumen enhancement in goats fed a diet based on rice straw without having an adverse effect on feed intake or nutrient digestion. However, further studies on the production of types of meat and milk that have a long-term feeding trial should be carried out.

## 1. Introduction

Animal nutritionists have a direct interest in modifying ruminal ecology and fermentation to enhance feed efficiency and reduce nitrogen excretion in ruminants. Feed additives that alter rumen fermentation promote feed utilization efficiency while lowering nitrogen excretion and protozoal population [1,2]. These effects were shown with the addition of antibiotics and ionophores, the majority of which are currently out of favor due to global concerns about food safety [2]. Plant phytonutrients are particularly appealing as rumen modifiers since they are natural substances [3]. Tannins and saponins are a group of phytonutrient compounds that have been demonstrated to alter rumen ecology, enhance rumen fermentation, increase dietary N utilization efficiency, and reduce N losses in ruminant animals [4].

*Terminalia chebula* Retz. (Combretaceae family) is a plant native to India and Southeast Asia whose fruits are a rich source of secondary compounds such as condensed tannins (CT) and saponins (SP) [5,6,7]. Our earlier studies showed that adding *Terminalia*
*chebula* meal (TCM) at 24 mg/g DM was shown to increase rumen fermentation and decrease protozoal population in vitro [6]. According to Patra et al. [8], supplementing 0.50 mL/g DM of TCM extract showed a reduction in the total protozoal population. In addition, Patra et al. [9] indicated that the addition of TCM at 10 g/kg of DM intake could increase fiber digestibility without adversely affecting DM intake and N utilization in sheep. Patra and Saxena [10], tannins, and saponins reduce the protozoa associated with fibrolytic bacteria [11]. Plant secondary metabolites have been shown to enhance ruminant feed intake, digestibility, and rumen fermentation characteristics [12]. This outcome may be due to protozoa, which are able to digest bacterial and fungal cells; as a result, reducing these protozoa is anticipated to increase the density of the rumen microbial population [11], which can improve digestibility and rumen fermentation in ruminants. However, the utilization of CT and SP in TCM for goats has not yet been studied. Based on the research on TCM as a feed additive in ruminants, we hypothesized that TCM could enhance rumen fermentation and nitrogen utilization, while resulting in a reduction in protozoal numbers in goats.

Therefore, the aim of the current experiment was to investigate the effects of TCM supplementation on the digestibility of nutrients and on rumen fermentation, N utilization, and protozoal population in goats fed a rice straw–based diet.

## 2. Materials and Methods

### 2.1. Animals, Diets, and Experimental Design

Fresh *Terminalia chebula* fruits were taken in the provinces of Sakon Nakhon and Udon Thani, Thailand. The fruits were dried at 60 °C for approximately 2 days and ground to create TCM. Eight male crossbred (Anglo-Nubian x Thai native) goats with live weights of 13 ± 3 kg of BW were arranged in a four-by-four replicated Latin square design. The TCM was supplemented at 0, 8, 16, and 24 g/kg of total DM intake. The goats were fed the concentrate diets (Table 1) at 13 g/kg BW and free-choice feedings of rice straw at 07:00 and 16:00. Each goat was kept separately in ventilated pens with access to clean, fresh water and mineral blocks at all times. The research was conducted over four periods of 21 days each. The first 14 days were used to measure feed intake, and the final 7 days were used to collect urine and feces. The BW of each goat was measured at the first and last day of each period.

### 2.2. Data Collection and Chemical Analysis

During the last week of the collection period, samples of feed, feces, and urine were collected from each goat. The volumes of DM, ash, CP [13], NDF, and ADF [14] in the samples were measured. The modified vanillin-HCl technique based on Burns [15] was used to analyze the condensed tannin content in the TCM. The crude saponins were determined using methanol extraction as described by Kwon et al. [16] and as modified by Poungchompu et al. [17]. A plastic container treated with sulfuric acid (10%) to prevent N loss was used to collect all of the urine samples; then they were evaluated for total N [18].

On the last day of each period, blood samples were taken from the jugular vein at 0 and 4 h after feeding to analyze blood urea nitrogen (BUN) [19]. Rumen fluid (approximately 60 mL) was collected from all the animals along with blood samples using a stomach tube linked to a vacuum pump. Ruminal pH was immediately measured using a mobile pH meter (FiveGO; Mettler-Toledo GmbH, Greifensee, Switzerland). Rumen fluid samples were filtered through four layers of cheesecloth and subsampled for NH_3_-N measurement and microbial counts. The concentration of NH_3_-N was determined using a Kjeltech Auto 1030 Analyzer [18], and volatile fatty acids (VFA) were quantified using high-pressure liquid chromatography [20]. The total direct counts of bacteria, protozoa, and fungi were obtained using the Galyean method [21].

### 2.3. Statistical Analysis

Using the GLM model approach in the SAS program, all the data were submitted to variance analysis using a repeated 4 × 4 Latin square design [22]. Data were analyzed using the model Yijk = μ + Mi + Aj + Pk + εijk, where Yijk is the observation from treatment i, goat j, and period k; μ is the overall mean; Mi is the mean effect of the treatments (I = 1–4); Aj is the mean effect of the goats (j = 1–8); Pk is the mean effect of the periods (k = 1–4); and εijk is the residual error. To evaluate the impact of TCM supplementation, Tukey’s test was utilized and *p* < 0.05 was used to establish whether the influence was significant.

## 3. Results

### 3.1. Feed Intake and Nutrient Digestibility

The addition of TCM did not affect the roughage, concentrate, and total DM intake (*p* > 0.05) (Table 2). Similarly, the intakes of nutrients were similar among the groups (*p* > 0.05). However, goats receiving TCM at 24 g/kg of total DM intake had a lower apparent digestibility of CP as compared to other treatments (*p* < 0.05). The CP digestibility was reduced from 60.6 to 56.1% with the increasing level of TCM.

### 3.2. Rumen Ecology, Microorganisms, and Blood Metabolites

The supplementation of TCM did not change the ruminal pH and BUN concentrations (*p* > 0.05) (Table 3). On the other hand, ruminal NH_3_-N at 4-h post feeding decreased in goats receiving TCM at 16 and 24 g/kg of total DM intake (*p* < 0.05). The NH_3_-N concentration was reduced from 18.9 to 17.0 mg/dl with the increasing level of TCM. In contrast, the total bacteria counts at 4-h post feeding increased with the inclusion of TCM as compared to the control group (*p* < 0.05). Total protozoa, horotrich, and entodiniomorph were reduced when supplemented with TCM (*p* < 0.05), while fungal zoospore counts were similar among the treatments (*p* > 0.05).

### 3.3. Volatile Fatty Acid (VFA) Profiles

Total VFA, acetic acid, and butyric acid concentrations and acetic/propionic ratio were unaffected by the addition of TCM (*p* > 0.05; Table 4). However, there was the exception that after 4-h post feeding, the concentration of propionic acid was higher for goats receiving TCM at 16 g/kg of total DM intake than for goats in the other treatment groups (*p* < 0.05).

### 3.4. Nitrogen Utilization

The use of TCM for goats had no effect on N intake or total N excretion (*p* > 0.05) (Table 5). However, the amount of N excreted in the urine decreased (*p* < 0.05), while fecal N excretion increased with the TCM-supplemented groups. In addition, N absorption, N retention, and the ratio of N retention to N intake were enhanced (*p* < 0.05) in goats fed with TCM at 8 and 16 g/kg of total DM intake.

## 4. Discussion

### 4.1. Feed Intake and Nutrient Digestibility

The supplementation of plants containing CT and SP to ruminant diets at a high level (>50 g/kg DM) usually reduces feed intake due to a decrease in palatability, a decrease in digestion rate, and the development of conditioned aversion [23,24]. In contrast, low to moderate tannin supplementation (<50 g/kg DM) has positive effects such as reducing bloat, inhibiting parasite larvae, and protecting protein digestion in the rumen [25,26,27]. In the present study, TCM was an alternative animal feed that did not receive a negative response regarding its palatability, as there were no negative effects on feed intake and nutrient intake. Similarly, Gunun et al. [28] found that CT-containing seed meal from *Antidesma thwaitesianum* Muell. Arg. could be utilized as a feed additive without causing negative effects on dairy cow feed intake.

The CP digestibility was lower with TCM supplementation. A decrease in CP digestibility was also observed by Tseu et al. [29], who included tannins in Nellore cows’ diets. Patra et al. [9] also found that the addition of tannin sources from TCM decreased ruminal apparent digestibility. The reduction of CP digestibility may have resulted from protein tannin complex formation, which may have reduced protein solubility and may have contributed to the decrease in CP digestibility [29]. This action is regarded as a positive impact of tannin ingestion. When feed protein is not degraded to ammonia in the rumen, the feed protein flow in the small intestine will increase, so that high-quality protein from feed can be absorbed from the small intestine [30]. This is in agreement with Barry and Manley [26], who found that the CT from *Lotus pedunculatus* can improve the post-ruminal flow of nitrogen and amino acids, thus increasing the level of the rumen bypass protein.

### 4.2. Rumen Ecology, Microorganisms, and Blood Metabolites

The current experiment has shown that ruminal pH did not differ among treatments, and the values were in an optimal range for normal rumen fermentation, microbial growth, and microbial activity [31]. In addition, ruminal NH_3_-N concentration at 4-h post feeding significantly decreased (*p* < 0.05) with the increasing level of TCM supplementation. This may be a result of the CT forming a protein tannin complexation, which reduces the availability of feed protein breakdown in the rumen and NH_3_-N release [32]. Moreover, BUN is a marker for rumen protein breakdown [33]. Although CP digestibility was reduced by diet supplementation, BUN concentration was not affected by TCM supplementation. In the present study, BUN was in the normal range of 10 to 15 mg/dl [34].

The total protozoa, horotrich, and entodiniomorph populations were reduced when supplemented with TCM containing CT and SP. Wallace et al. [35] explained that SP might kill or damage protozoa by reacting with the cholesterol contained in the membrane of the protozoa. The membrane may degrade and eventually disintegrate. The growth or activities of methanogens and protozoa in the rumen may also be inhibited by tannins [36]. In the previous in vitro trial, adding TCM at 20 mg/0.5 g of DM reduced the protozoal count by 68% [6]. Likewise, Patra et al. [8] reported that total protozoa counts were reduced with the addition of TCM extract at 0.50 mL/g DM. Moreover, the total bacteria population increased when supplemented with TCM, possibly as a result of the suppression of protozoal populations. Meanwhile, the TCM did not affect the fungal zoospores.

### 4.3. Volatile Fatty Acid (VFA) Profiles

An increase in the proportion of propionic acid was observed. Patra and Saxena [37] suggest that SP may also induce a shift in the production of propionate from excess hydrogen. The expected shift in the VFA profile from the acetic to propionic acid ratio was correlated with a shift in excess hydrogen, which is particularly helpful for producing propionic acid [6]. As a result, the reducing power must be eliminated by an alternative metabolic pathway that uses an electron sink. The synthesis of propionic acid through the succinate propionic acid pathway is a possible means to achieve this [38]. Notably, when goats were fed TCM as a source of CT and SP, the proportion of propionic acid increased.

### 4.4. Nitrogen Utilization

High protein consumption and rapid ruminal digestion cause ammonia production to exceed microbial needs, which results in high urinary N. Extra ammonia can be converted to urea in the liver and recycled through the ruminal wall, salivary secretion, and urine excretion. In the present study, supplementation with TCM as a source of CT was shown to reduce NH_3_-N and urinary N excretion, whereas fecal N excretion increased as compared to the control group. These results can be explained by the fact that tannins may inhibit protein degradation, due to the formation of tannin protein complexes that inhibit protein breakdown and which may disrupt post-rumen enzymatic activity, leading to an increase in fecal N [39]. High-quality protein is more efficiently absorbed in the lower gut due to its condensed tannins and saponins [31]. The addition of TCM enhanced N absorption and retention and the proportion of N retention to N intake. This was possible because the animals retained protein to make up for natural protein losses, such as the enhanced secretion of salivary glycoproteins and digestive enzymes; greater epithelial cell regeneration; and higher mucus secretion in the gastrointestinal tract with the addition of tannins [39]. Similar findings were reported by Dawson et al. [40]: Adding Quebracho tannin extract at 5% to sheep promoted surface epithelial ulceration and enhanced mucosal histiocytes in the jejunum and ileum.

## 5. Conclusions and Recommendations

The TCM supplementation improved rumen fermentation and N balance without affecting feed intake and nutrient digestibility. The TCM supplementation could also reduce the protozoal population. This study recommends the dosed usage of TCM at 16 g/kg of total DM intake as a rumen modifier.

## Figures and Tables

**Table 1 animals-12-02022-t001:** Ingredients and chemical compositions of concentrate, rice straw, and *Terminalia chebula* meal (TCM).

Item	Concentrate	Rice Straw	TCM
Ingredient, g/kg dry matter (DM)			
Rice straw	400		
Cassava chip	300		
Rice bran	100		
Soybean meal	140		
Urea	20		
Molasses	20		
Minerals and vitamins	10		
Pure sulfur	5		
Salt	5		
Chemical composition			
Dry matter, g/kg	890	910	944
Organic matter, g/kg DM	929	863	969
Crude protein, g/kg DM	140	33	38
Neutral detergent fiber, g/kg DM	356	697	327
Acid detergent fiber, g/kg DM	222	576	201
Condensed tannin, g/kg DM	-	-	81
Crude saponin, g/kg DM	-	-	94

**Table 2 animals-12-02022-t002:** Effect of *Terminalia chebula* meal (TCM) supplementation on feed intake and digestibility.

Items	TCM (g/kg of Total DM Intake)				SEM	*p*-Value
	0	8	16	24		
Roughage DM intake						
g/d	244.2	241.0	246.8	238.3	1.62	0.27
g/kg BW	14.2	14.3	14.0	13.9	0.03	0.20
g/kg BW^0.75^	274.6	265.5	268.8	265.1	0.61	0.37
Concentrate DM intake						
g/d	229.2	221.1	216.8	223.6	2.41	0.78
g/kg BW	13.0	13.1	13.0	13.1	0.16	0.21
g/kg BW^0.75^	251.3	256.2	257.8	259.3	0.17	0.27
Total DM intake						
g/d	473.3	462.1	463.7	461.8	2.64	0.85
g/kg BW	27.3	27.4	27.1	26.9	0.04	0.10
g/kg BW^0.75^	525.9	511.0	512.2	523.8	1.16	0.23
Nutrients intake, g/d						
Dry matter	473.3	462.1	463.6	461.8	2.94	0.73
Organic matter	391.9	391.3	378.7	379.1	2.94	0.60
Crude protein	40.2	40.4	39.9	40.7	0.41	0.41
Neutral detergent fiber	244.1	241.24	242.88	241.4	2.47	0.34
Acid detergent fiber	184.0	181.1	181.7	181.3	1.66	0.97
Apparent Digestibility, %						
Dry matter	61.0	64.9	61.7	61.1	1.51	0.78
Organic matter	62.8	67.5	63.3	63.2	1.95	0.72
Crude protein	60.6 ^a^	59.9 ^a^	58.7 ^ab^	56.1 ^b^	1.10	0.04
Neutral detergent fiber	58.0	57.8	56.4	56.0	0.88	0.61
Acid detergent fiber	57.1	56.9	56.0	55.9	0.84	0.81

^a,b^ Values on the same row with different superscripts differed (*p* < 0.05).

**Table 3 animals-12-02022-t003:** Effect of *Terminalia chebula* meal (TCM) supplementation on ruminal pH, NH_3_-N, BUN concentration, and microbial population.

Items	TCM (g/kg of Total DM Intake)				SEM	*p*-Value
	0	8	16	24		
Ruminal pH						
0 h-post feeding	6.7	6.8	6.6	6.7	0.06	0.89
4 h-post feeding	6.5	6.5	6.6	6.5	0.04	0.81
mean	6.6	6.6	6.6	6.6	0.02	0.40
NH_3_-N concentration, mg/dL						
0 h-post feeding	14.6	15.2	14.2	14.9	0.61	0.44
4 h-post feeding	18.8 ^a^	18.9 ^a^	17.0 ^b^	17.0 ^b^	0.31	0.04
mean	16.7	17.0	15.7	16.0	0.39	0.41
Blood urea-N concentration, mg/dL						
0 h-post feeding	13.0	14.0	12.9	12.4	0.56	0.75
4 h-post feeding	14.4	15.0	13.5	14.5	0.51	0.88
mean	13.7	14.5	13.2	13.4	0.55	0.65
Direct count, (cell/mL)						
Total bacteria, ×10^8^						
0 h post feeding	4.3	4.2	4.3	4.3	0.30	0.26
4 h-post feeding	4.1 ^b^	5.4 ^a^	5.5 ^a^	5.7 ^a^	0.21	0.02
Mean	4.2	4.8	4.9	5.0	0.31	0.16
Protozoa, ×10^5^						
Entodiniomorph						
0 h post feeding	5.3	4.5	4.3	4.5	0.43	0.22
4 h-post feeding	6.1 ^a^	4.7 ^b^	4.4 ^b^	4.1 ^b^	0.51	<0.01
Mean	6.0 ^a^	4.8 ^b^	4.4 ^b^	4.4 ^b^	0.34	0.03
Holotrich						
0 h post feeding	4.1	3.4	3.3	3.4	0.33	0.29
4 h-post feeding	6.9 ^a^	4.2 ^b^	3.9 ^b^	3.7 ^b^	0.52	<0.01
Mean	4.7 ^a^	3.7 ^b^	3.6 ^b^	3.7 ^b^	0.33	0.04
Total Protozoa						
0 h post feeding	9.4	7.9	7.7	7.9	0.64	0.51
4 h-post feeding	13.0 ^a^	8.9 ^b^	8.3 ^b^	7.8 ^b^	0.78	<0.01
Mean	10.7 ^a^	8.4 ^b^	8.0 ^b^	8.0 ^b^	0.61	0.03
Fungi zoospore, ×10^4^						
0 h post feeding	6.0	6.2	6.2	6.0	0.23	0.20
4 h-post feeding	6.3	6.6	6.6	7.6	0.31	0.44
Mean	6.2	6.4	6.4	6.7	0.26	0.33

^a,b^ Values on the same row with different superscripts differed (*p* < 0.05).

**Table 4 animals-12-02022-t004:** Effect of supplementing meals with *Terminalia chebula* meal (TCM) on ruminal volatile fatty acid profile.

Items	TCM (g/kg of Total DM Intake)				SEM	*p*-Value
	0	8	16	24		
Total VFA, mmol/L						
0 h-post feeding	127.5	139.9	144.2	129.3	4.44	0.21
4 h-post feeding	122.8	134.9	121.5	128.5	3.84	0.86
Mean	125.7	139.6	133.5	127.8	2.41	0.44
VFA, mol/100 mol						
Acetic acid						
0 h-post feeding	67.6	66.9	67.1	66.5	2.73	0.43
4 h-post feeding	67.8	62.5	64.8	65.9	2.11	0.08
Mean	67.7	64.7	65.9	66.2	2.00	0.11
Propionic acid						
0 h-post feeding	19.5	20.2	19.8	21.0	1.77	0.10
4 h-post feeding	21.9 ^b^	24.5 ^ab^	26.1 ^a^	24.9 ^ab^	0.67	0.02
Mean	20.7	22.4	23.0	23.0	1.21	0.11
Butyric acid						
0 h-post feeding	12.9	12.9	13.1	13.5	0.13	0.24
4 h-post feeding	10.3	13.0	9.1	9.2	0.79	0.08
Mean	11.6	13.0	11.1	12.9	1.42	0.20
Acetic/propionic acid ratio						
0 h-post feeding	3.5	3.3	3.4	3.0	0.59	0.21
4 h-post feeding	3.1	2.6	2.5	2.6	0.54	0.98
Mean	3.3	2.9	2.9	2.8	0.57	0.34

^a,b^ Values on the same row with different superscripts differed (*p* < 0.05).

**Table 5 animals-12-02022-t005:** Effect of *Terminalia chebula* meal (TCM) meal supplementation on N utilization.

Items	TCM (g/kg of Total DM Intake)				SEM	*p*-Value
	0	8	16	24		
N utilization						
N intake, g/d						
Roughage	1.1	1.2	1.2	1.0	0.06	0.65
Concentrate	5.5	5.3	5.3	5.4	0.05	0.70
Total	6.6	6.5	6.5	6.5	0.06	0.45
N excretion, g/d						
Feces	1.6 ^a^	2.2 ^b^	2.3 ^b^	2.2 ^b^	0.11	0.03
Urine	2.7 ^a^	1.6 ^b^	1.6 ^b^	1.7 ^b^	0.18	0.03
Total	4.3	3.8	3.9	3.9	0.20	0.41
Nitrogen balance g/d						
N absorption	4.1 ^b^	4.6 ^a^	4.5 ^a^	4.4 ^ab^	0.11	0.04
N retention	2.9 ^b^	3.6 ^a^	3.4 ^a^	2.9 ^b^	0.14	0.03
N retention/ N intake	49.9 ^c^	58.7 ^a^	57.4 ^a^	53.5 ^b^	0.19	0.04

^a,b,c^ Values on the same row with different superscripts differed (*p* < 0.05).

## Data Availability

Not applicable.

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
