# Peer review of "The Effect of Phytonutrients in Terminalia chebula Retz. on Rumen Fermentation Efficiency, Nitrogen Utilization, and Protozoal Population in Goats"

_animals, 2022, doi:10.3390/ani12162022_

Round 1

Reviewer 1 Report

find the file

Author Response

Reviewer 1

  1. Provide abberavation for Terminalia chebula Retz. At it first appearance and similar other issues sould be addressed.
  • Already update data, please see in the text.

  1. Similar studies have been conducted as mentioned in introduction part (Line: 73,74….. ), how authors justify the difference of this study from others.
  • They previously examined the impact of Terminalia chebula extract in an in vitro investigation, whereas we examined the effectiveness of Terminalia chebula meal in an in vivo study including feed intake, digestibility, rumen ecology and N utilization in goats.

  1. Part “3.2. Rumen ecology, microorganisms and blood metabolites” authors only reported general data such as bacteria, protozoa, etc, not specific species, did they identifies specific species?
  • In the present study, we use the total direct count method to estimate the population of microorganisms (Galyean, 1989). As a result, it is unable to identify specific species of bacteria.

  1. Normally VFA production increase with the food load, why it was heighst for 16g/kg, not for 24g/kg as shown in table 4?
  • Tannins and saponins are a group of phytonutrient elements in Terminalia chebula. Low to moderate concentrations of it in ruminant diets are considered to enhance rumen fermentation characteristics. Therefore, the suitable level of phytonutrients in Terminalia chebula meal, hence the high concentration of VFA. However, adding Terminalia chebula meal at 16 and 24 g/kg DM were not significantly different.

  1. In the results part most of the data are presented in table and very few are presented in text, I suggest you to explain more instead of just providing tables.
  • L142-143: Already added information, please see in the text.

In addition, is it possible to presented some results in Fig. form?

  • I am really sure that presenting the data as numbers in a table makes it easier for the reader to understand than presenting it in figures.

  1. Part “4.4. Nitrogen utilization” did the authors observed any adverse effect of NH3N on rumen microbes?
  • The concentration of NH3N ranges from 15.71 – 17.00 mg/dl which is an optimal range for normal rumen fermentation and microbial activity as observed from digestibility and fermentation end-product.

 “These results can be explained by the fact that tannins may inhibit protein degradation, due to the formation of tannin protein complexes, which inhibit protein breakdown, which may disrupt post-rumen enzymatic activity, leading to an increase in fecal N [40]” this reflects the drawback of this plant sp. In goat diet. What are the authors opinions?

  • The formation of tannin protein complexes inhibits protein degradation in the rumen while increasing by-pass protein to the lower gut, where the animal can directly digest protein and use it, leading to an increase in fecal N.

Reviewer 2 Report

The authors examined the effect of supplementing the condensed tannin- and saponin-containing Terminalia chebula meal to a rice-straw and concentrate diet fed to crossbred goats. In general, there were no effects on most metrics of feed intake and ruminal chemistry, thought there was a slight suppression of protozoal numbers and crude protein digestibility. This resulted in a more beneficial N balance, suggesting that supplementation with T. chebula meal may be beneficial in goat production. Overall, the experiments were adequately performed, with an acceptable level of replication, and the data adequately interpreted. The reviewer’s main concerns regards the use of an overly liberal means separation test; the lack of branched chain VFA data that would have bolstered observations on reduced protein digestibility; and the selection of 16 g/kg DM as the recommended supplementation level. These concerns are explained in greater detail below.

Specific comments:

L74: Express the value as a concentration, rather than as a volume.

L80-82: The problem with this argument is that the vast majority of fibrolytic bacteria are adherent to feed particles, and thus cannot be ingested by protozoa.

L86-87: How would enhancing rumen fermentation and nitrogen utilization cause a reduction in protozoal numbers?

Table 1: The values are expressed at an unreasonable level of precision.

L144: H2PO4, or H3PO4?

L157: Duncan’s New Multiple Range test is notoriously liberal. The authors should use a more conservative test, such as a Tukey test. The use of a more conservative test will not change the many nonsignificant effects observed, but it will provide a better test of the claimed significant effects.

L182-186: No data are provided for the branched-chain VFA. Such data would be useful, as these VFA are produced exclusively by fermentation of branched-chain AAs; the levels of these BCVFAs would be expected to differ among treatments if there were differences in protein digestibility (as was reported in Table 2).

L238: Twenty mg per what? The value should be expressed as a concentration.

L240: Again, express this as a concentration.

L245-253: This section is a bit unclear. Where does this “excess hydrogen” come from? Why should the amount of “excess hydrogen” differ among treatments?

L276: It’s not clear why the authors recommend a supplementation level of 16 g/kg DM. It seems that, at least based on protozoan suppression and improving N retention, that a supplementation level of 8 g/kg DM yields similar effects.

Minor edits:

L66: Change “elements” to “compounds”.

L105: Change “weighed” to “measured”.

L116: Change “by the goats” to “by goat”.

L125: Delete “A”.

L233: Change “between” to “among”.

L256-258: Rewrite as a complete sentence.

L259: Change “has been” to “was”.

L263-264: Change “absorbe” to “absorbed”.

Author Response

 Reviewer 2

Specific comments:

L74: Express the value as a concentration, rather than as a volume.

  • L74: Already changed to “0.50 mL/g DM of TCM extract”, please see in the text.

L80-82: The problem with this argument is that the vast majority of fibrolytic bacteria are adherent to feed particles, and thus cannot be ingested by protozoa.

  • L79-82: Already changed to “This outcome may be due to protozoa, which are able to digest bacterial and fungal cells; as a result, reducing these protozoa is anticipated to increase the density of rumen microbial population [11], please see in text.

L86-87: How would enhancing rumen fermentation and nitrogen utilization cause a reduction in protozoal numbers?

  • Several studies have shown that saponin-containing plants increase bacterial numbers, presumably as a consequence of inhibition of protozoal numbers (Hess et al., 2004; Diaz et al., 1993; Valdez et al., 1986; Thalib et al., 1995; Newbold et al., 1997). Therefore, reducing these protozoa is anticipated to increase the density of the rumen microbial population (Wanapat et al., 2012), which can improve digestibility and rumen fermentation in ruminants.

Table 1: The values are expressed at an unreasonable level of precision.

  • Already changed, please see in table 1.

L144: H2PO4, or H3PO4?

  • It is H2PO4.

L157: Duncan’s New Multiple Range test is notoriously liberal. The authors should use a more conservative test, such as a Tukey test. The use of a more conservative test will not change the many nonsignificant effects observed, but it will provide a better test of the claimed significant effects.

  • L128: We used a Tukey test of statistical analysis, please see in text.

L182-186: No data are provided for the branched-chain VFA. Such data would be useful, as these VFA are produced exclusively by fermentation of branched-chain AAs; the levels of these BCVFAs would be expected to differ among treatments if there were differences in protein digestibility (as was reported in Table 2).

  • In current study we do not analyze for branched-chain VFA. However, an increasing of propionic acid was indirect effect of tannin containing Terminalia chebula

L238: Twenty mg per what? The value should be expressed as a concentration.

  • L202-203: Already changed to “In the previous in vitro trial, adding TCM at 20 mg/0.5 g of DM…….”, please see in the text.

L240: Again, express this as a concentration.

  • L205: Already changed to “…….TCM extract at 0.50 mL/g DM.”, please see in the text.

L245-253: This section is a bit unclear. Where does this “excess hydrogen” come from? Why should the amount of “excess hydrogen” differ among treatments?

  • We have been discuss for the expected shift in the VFA profile from the acetic to propionic acid ratio was correlated with a shift in hydrogen from the CH4 pathway, which is particularly helpful for producing propionic acid. But editor request us remove all data and relative information about CH4.

L276: It’s not clear why the authors recommend a supplementation level of 16 g/kg DM. It seems that, at least based on protozoan suppression and improving N retention, that a supplementation level of 8 g/kg DM yields similar effects.

  • We recommend a level of 16 g/kg DM because this level shows an increase in propionic acid, N utilization while reducing CP digestibility and protozoa population.

Minor edits:

L66: Change “elements” to “compounds”.

  • L66: Already changed, please see in text.

L105: Change “weighed” to “measured”.

  • L101: Already changed, please see in text.

L116: Change “by the goats” to “by goat”.

  • Already changed the sentences as the suggestion by editors, due to high duplication rate, especially materials and methods, please see in text.

L125: Delete “A”.

  • Already changed the sentences as the suggestion by editors, due to high duplication rate, especially materials and methods, please see in text.

L233: Change “between” to “among”.

  • L189: Already changed, please see in text.

L256-258: Rewrite as a complete sentence.

  • L219-221: Already changed to “Extra ammonia can be converted to urea in the liver and recycled through the ruminal wall, salivary secretion, and urine excretion.”, please see in text.

L259: Change “has been” to “was”.

  • L221: Already changed, please see in text.

L263-264: Change “absorbe” to “absorbed”.

  • L226: Already changed, please see in text.
